



# Continuous temperature soundings at the stratosphere and lower mesosphere with a ground-based radiometer considering the Zeeman effect

Witali Krochin[1], Francisco Navas-Guzmán[2,3], David Kuhl[4], Axel Murk[1], and Gunter Stober[1]

[1]University of Bern & Oeschger Center for Climate Change Research, Bern
[2]University of Granada, Andalusian Institute for Earth System Research, Granada, Spain
[3]Federal Office of Meteorology and Climatology MeteoSwiss, Payerne, 1530, Switzerland
[4]Naval Research Laboratory, Washington DC, USA

**Correspondence:** Witali Krochin (witali.krochin@unibe.ch)

**Abstract.** Continuous temperature observations at the stratosphere and lower mesosphere are rare. Radiometry opens the possibility by observing microwave emissions from two oxygen lines to retrieve temperature profiles at all altitudes. In this study, we present observations performed with a temperature radiometer (TEMPERA) at the Meteoswiss station at Payerne for the period from 2014 to 2017. We reanalyzed these observations with a recently developed and improved retrieval algorithm accounting for the Zeeman line splitting in the line center of both oxygen emission lines at 52.5424 and 53.0669 GHz. The new temperature retrievals were validated against MERRA2 reanalysis and the meteorological analysis NAVGEM-HA. The comparison confirmed that the new algorithm yields an increased measurement response up to an altitude of 53-55 km, which extends the altitude coverage by 8-10 km compared to previous retrievals without considering the Zeeman effect. Furthermore, we found correlation coefficients comparing the TEMPERA temperatures with MERRA2 and NAVGEM-HA for monthly mean profiles to be in the range of 0.8-0.96. In addition, mean temperature biases of 1 K and -2 K were found between TEMPERA and both models (MERRA2 and NAVGEM-HA), respectively. We also identified systematic altitude-dependent cold and warm biases compared to both model data sets.

## 1 Introduction

Continuous temperature soundings with high temporal and vertical resolution at the stratosphere and lower mesosphere are experimentally challenging, but desirable to measure continuously the temperature at the stratosphere/lower mesosphere and to assess the intermittent behaviour of atmospheric waves and understanding the day-to-day variability of the forcing from below in the ionosphere and thermosphere for space weather applications (Liu, 2016). Continuous observations of atmospheric temperature in the middle atmosphere are crucial to understand the chemistry (e.g., ozone) (Stolarski et al., 2012; Anderson et al., 2017) and to infer dynamics due to the thermal wind balance (Matthias and Ern, 2018).

Satellite observations provide global coverage. SABER (Sounding of the Atmosphere using Broadband Emission Radiometry) on board the TIMED (Thermosphere-Ionosphere-Mesosphere-Energy and Dynamics) satellite measures temperatures from the troposphere up to mesosphere/lower thermosphere. The satellite has an orbit around Earth that permits to cover all local





times within 60-days and, thus, provides only limited information on the short-term variability of tides and planetary waves. Furthermore, the latitudinal coverage changes in time due to the yaw cycle of the spacecraft (Russell III et al., 1999; Remsberg

et al., 2008; Rezac et al., 2015). The Microwave Limb Sounder (MLS) (Waters et al., 2006) on the AURA satellite (Schoeberl et al., 2006) is on a sun-synchronous orbit and, thus, passes at fixed local times the same geographic locations making a data analysis of tides and their intermittency unfeasible, although MLS obtains temperatures from the stratosphere up to mesosphere covering all latitudes between 82° N and 82° S.

However, for low- and mid-latitudes SABER observations have been utilized to gain some insight into the climatological sea-

sonal behaviour of the migrating and non-migrating diurnal and semidiurnal tides (Oberheide et al., 2011; Dhadly et al., 2018). Furthermore, these satellite observations have been proven to be valuable for data assimilation purposes into General Circulation Models (GCMs) such as the Navy Global Environment Model - High Altitude (NAVGEM-HA) (Eckermann et al., 2018). NAVGEM-HA temperature and wind fields show reasonable agreement to ground-based observations and the underlying day-to-day variability due to atmospheric tides and planetary waves (McCormack et al., 2017; Stober et al., 2020).

Continuous ground-based temperature observations of the stratosphere and mesosphere are challenging and ambitious. There are only a few Rayleigh lidar measurements that are long enough to infer the tidal variability (Baumgarten et al., 2018; Baumgarten and Stober, 2019). Mainly due to the fact that lidar observations are weather dependent, which essentially limits the measurement time and data availability. Furthermore, some of these lidars have only nighttime capabilities (Wing et al., 2018; Sica and Haefele, 2015) introducing additional ambiguities to infer mean temperatures and to assess the tidal variability.

Microwave radiometry offers a robust remote sensing technique that is almost weather independent to retrieve atmospheric temperature profiles at the stratosphere and lower mesosphere. A few years ago the University of Bern developed a temperature radiometer TEMPERA (TEMPErature RAdiometer) to perform continuous soundings including the troposphere (Stähli et al., 2013; Navas-Guzmán et al., 2016). Recently, we developed a new retrieval algorithm due to updates in the radiative transfer model ARTS (Buehler et al., 2018; Eriksson et al., 2005) and revised Quantum numbers of HITRAN. The new retrieval

algorithm accounts for the Zeeman effect at the line center in both emission lines at 52.5424 and 53.0669 GHz for routine temperature soundings. The advantage of the new retrieval algorithm is an increased altitude coverage. In this study, we present a validation of the new temperature profiles against MERRA2 and NAVGEM-HA for the location Payerne in Switzerland.

The manuscript is structured as follows. Section 2 contains a brief description of the temperature radiometer TEMPERA and section 3 summarizes the Zeeman effect on the oxygen emission lines. MERRA2 and NAVGEM-HA data sets are presented in

section 4. The retrieval algorithm is outlined in section 5. The TEMPERA temperature soundings and validation are shown in sections 6 and 7. The results are discussed in section 8. Our conclusions are summarized in section 9.

## 2  The TEMPERA radiometer

TEMPERA is a ground-based radiometer developed at the University of Bern. It measures atmospheric microwave radiation in the range of the Oxygen emission complex at 50-60 GHz. For stratospheric temperature retrievals, two emission lines of the

$O_2$ molecule are observed with a high-resolution digital FFT spectrometer at 52.5424 GHz and 53.0669 GHz with a resolution





of 30.5 kHz and a bandwidth of 960 MHz. The instrument was located at the aerological station in Payerne (46.82N, 6.95°E, 491 m asl) and was directed westwards with an elevation angle of 60°. The antenna half-beam-width (HPBW) is 4°. A more detailed technical description of the instrument can be found in Stähli et al. (2013). The measured spectra can be inverted into vertically resolved temperature profiles considering the pressure broadening of the spectral emission lines and their radiative

transfer. Retrievals presented in this study make use of the Atmospheric Radiative Transfer Simulator (ARTS) (Buehler et al., 2018) and Qpack, the Matlab interface, for ARTS (Eriksson et al., 2005).

Already in 2015 first observations of the Zeeman effect in the line center for atmospheric Oxygen were reported using TEMPERA (Navas-Guzmán et al., 2015). In 2017 Navas-Guzmán et al. (2017) presented a comparison of almost three years of continuous TEMPERA observations with radiosondes, the Microwave Limb Sounder (MLS) on board the AURA spacecraft

and a Rayleigh lidar. These former studies inferred stratospheric temperature profiles up to an altitude of 40-45 km altitude blanking the line center to avoid a contamination of the temperature measurements due to the Zeeman line broadening, which was not included at this time in the retrievals due to limitations in the available databases for the radiative transfer and quantum numbers in HITRAN that are required to account for the Zeeman effect in both oxygen lines (Larsson et al., 2019).

The observations presented in this study were performed with the laboratory prototype between 2014-2017 (Stähli et al., 2013).

The receiver was upgraded in July 2015, which improved the overall performance of the instrument. The upgrade showed much better suppression of the standing waves. However, the new receiver introduced a small temperature offset in the calibrated and tropospheric corrected spectra of about 0.6 K.

## 3  The Zeeman effect

The Zeeman effect is a splitting of energy levels in emission and absorption processes due to an interaction of the involved

molecules with a magnetic field. Atmospheric oxygen has a permanent magnetic moment that interacts with Earth's magnetic field. Therefore an emission line, coming from rotational transitions, splits up into several lines. The degree of the line splitting depends on the strength of the magnetic field. Earth's magnetic field is rather weak, compared to stellar magnetic fields often analyzed in astronomy, which leads more to a broadening of the line center rather than a visible separation of individual Zeeman lines for each energy level. At mesospheric altitudes where the atmospheric pressure is already low, Zeeman broadening

dominates over pressure broadening. Thus, temperature retrievals above 45 km are no longer feasible without taking into account the Zeeman effect. The change of the line shape due to the Earth's magnetic field for both frequencies is demonstrated in Figure 1. The importance of the magnetic field is evident from these theoretical spectra. The new retrieval algorithm (Larsson et al., 2019) considers the Zeeman effect for both oxygen emission lines.



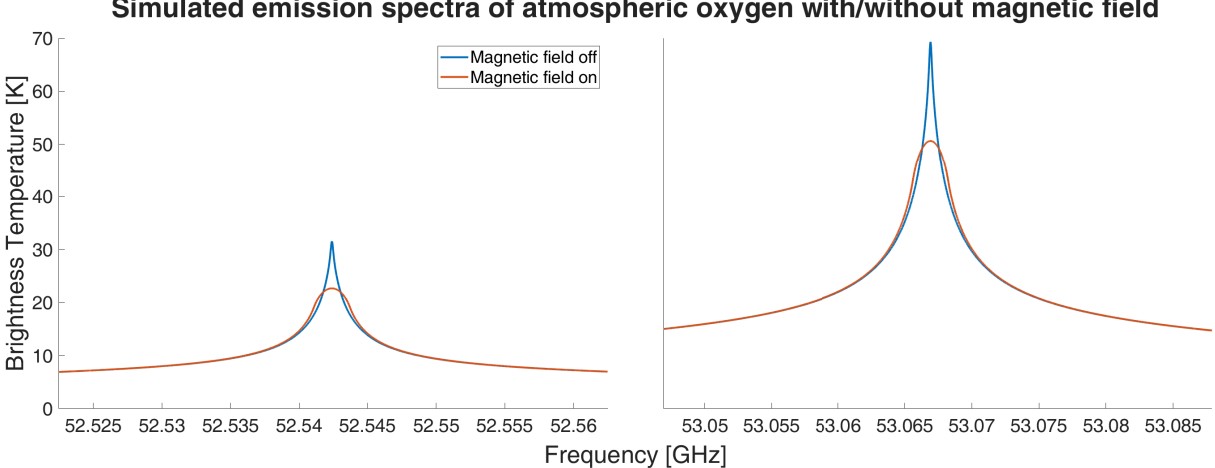

**Figure 1.** Illustration of the Zeeman effect on the line shape for mid-latitude observations on Earth. The line of sight is directed northwards with a zenith angle of $30°$. The tropospheric effect on the brightness temperature has been removed.

## 4 MERRA2 and NAVGEM-HA

Stratospheric and mesospheric temperatures obtained from the new retrieval algorithm are compared to MERRA2 reanalysis (Gelaro et al., 2017) and to the meteorological analysis NAVGEM-HA (Eckermann et al., 2018). The vertical temperature profiles are extracted for the location of Payerne considering the spatial averaging of the radiometer of about 250 km in diameter keeping the temporal resolution of the model fields of 3 hours. Only the vertical resolution of the model data was interpolated to a fixed altitude grid with 2 kilometer vertical resolution to simplify the comparisons. MERRA2 reanalysis utilizes a 3DVAR assimilation tool (e.g. Gelaro et al., 2017, and references therein), which updates the state vector every 6 hours. A detailed description of the Hybrid 4-DVAR data assimilation in NAVGEM-HA is provided in Kuhl et al. (2013) and Eckermann et al. (2018). Similar to MERRA2 the model state vector is updated every 6 hours at the mesosphere.

For the comparison with the temperature observations from TEMPERA, the model data was analyzed at the geographic location of Payerne and all grid points in a 250 km radius were averaged after they had been interpolated to a geometric vertical altitude grid. Daily mean temperatures and tidal amplitudes were derived by an adaptive spectral filter similar to Pokhotelov et al. (2018); Baumgarten and Stober (2019); Stober et al. (2020). The geopotential altitudes from NAVGEM-HA were converted into geometric heights (Stober et al., 2021). The temporal resolution of 3 hours for both model data were kept.





## 5 Temperature retrieval with optimal estimation

### 5.1 Temperature retrievals

The inversion of the forward model is solved with ARTS 2.4 (Atmospheric Radiative Transfer Simulator, Buehler et al. (2018)). The mathematical method follows the formalism from Rodgers (2000) and is briefly explained in this section.

Consider $\boldsymbol{y}$ as the measurement vector and $\boldsymbol{x}$ as the state vector. In our case $\boldsymbol{y}$ is the spectrum with $n$ channels and $\boldsymbol{x}$ is the
temperature profile with $m$ grid points. The forward model $F(\boldsymbol{x}, \boldsymbol{b})$ maps the atmospheric state $\boldsymbol{x}$ to an idealized spectrum, this is usually written as;

$$\boldsymbol{y} = F(\boldsymbol{x}, \boldsymbol{b}) + \boldsymbol{\epsilon} \ . \tag{1}$$

The vector $\boldsymbol{b}$ contains some other parameters that are not included in the state vector and $\boldsymbol{\epsilon}$ is the measurement error. The challenge is to find an inversion of the forward model $F(\boldsymbol{x}, \boldsymbol{b})$ that presents an optimal estimate to the observations. The
problem is that there is often no unique state $\boldsymbol{x}$ for a given measurement $\boldsymbol{y}$, which is classified as ill-posed. The inversion (also called retrieval) can be understood as a mapping $R$ of the measurement vector $\boldsymbol{y}$ onto an optimal state vector $\hat{\boldsymbol{x}}$;

$$\hat{\boldsymbol{x}} = R(\boldsymbol{y}, \hat{b}, \boldsymbol{x}_a, \boldsymbol{c}) \ , \tag{2}$$

here $\hat{b}$ is the best estimate of the forward model parameters, $\boldsymbol{x}_a$ denotes the apriori knowledge on the state vector and $\boldsymbol{c}$ are some additional parameters. The Optimal Estimation Method (OEM) provides the most probable solution $\hat{\boldsymbol{x}}$ in the context of
the forward model. To apply this method information about the atmospheric state must be added. This information is included in the apriori state $\boldsymbol{x}_a$, which is a pre-knowledge background state of the atmosphere. The choice of a certain apriori is crucial and is explained in section 5.2. The error-covariance of the apriori state is described in the apriori-covariance matrix $\mathbf{S}_a$, and the measurements errors are described in the measurement-error covariance matrix $\mathbf{S}_\epsilon$. The optimal solution can be found by maximising the probability $P(\boldsymbol{x}|\boldsymbol{y})$ of $\boldsymbol{x}$ under the condition that $\boldsymbol{y}$ is known or in this case equivalent and most common,
minimising the cost function $J(\boldsymbol{x}) = -2\ln P(\boldsymbol{x}|\boldsymbol{y})$, which can be written in the form;

$$J(\boldsymbol{x}) = [\boldsymbol{y} - F(\boldsymbol{x})]^T \mathbf{S}_\epsilon^{-1} [\boldsymbol{y} - F(\boldsymbol{x})] + [\boldsymbol{x} - \boldsymbol{x}_a]^T \mathbf{S}_a^{-1} [\boldsymbol{x} - \boldsymbol{x}_a] \ . \tag{3}$$

The derivation of this cost function is based on the Bayes' probability theorem and the assumption that the probability distributions for the apriori covariance $\mathbf{S}_a$ and for $\mathbf{S}_\epsilon$ and as well as the posterior distribution $\boldsymbol{x}$ are Gaussian. The minimum of $J(\boldsymbol{x})$ is found by the following condition;

$$\nabla_{\boldsymbol{x}} J(\boldsymbol{x}) = 0 \ . \tag{4}$$

This equation is solved using several iterations making use of the Levenberg-Marquardt solver. Thus, successive iterations are computed from;

$$\boldsymbol{x}_{i+1} = \boldsymbol{x}_i + \left(\mathbf{S}_a^{-1} + \mathbf{K}_i^T \mathbf{S}_\epsilon^{-1} \mathbf{K}_i + \gamma D\right)^{-1} \left[\mathbf{K}_i^T \mathbf{S}_\epsilon^{-1} (\boldsymbol{y} - F(\boldsymbol{x})) - \mathbf{S}_a^{-1} (\boldsymbol{x} - \boldsymbol{x}_a)\right] \ , \tag{5}$$



where $\mathbf{K} = \partial F / \partial \boldsymbol{x}$ is called the weighting function. The apriori profile was used for the 0-th step $\boldsymbol{x}_0 = \boldsymbol{x}_a$. For $\gamma = 0$ this
method is equivalent to the Gauss-Newton method. The damping term $\gamma D$ ensures the iteration to converge, even under poor
conditions, making this method more robust but also slower compared to the Gauss-Newton scheme.

## 5.2    Apriori atmospheric information

The retrieval algorithm is initialized using an ECMWF climatology. The climatology was obtained averaging daily mean
ECMWF data between 2014-2017 smoothed by a 30 day running window. The resulting seasonal apriori temperature be-
haviour is shown in Figure 2. Based on this climatological mean atmospheric state, the radiative transfer equations are solved
for several molecular species e.g., $O_2$, $H_2O$, $O_3$, and $N_2$. Although not all of them contribute significantly to the radiation
intensity at 50-60 GHz. Spectroscopic data for $O_2$ was taken from the HITRAN database (Gordon et al., 2017). These quantum
numbers are necessary to account for the Zeeman effect in the radiative transfer model. The magnetic field strength for the
location of Bern at the altitude of the mesosphere is taken from ARTS.

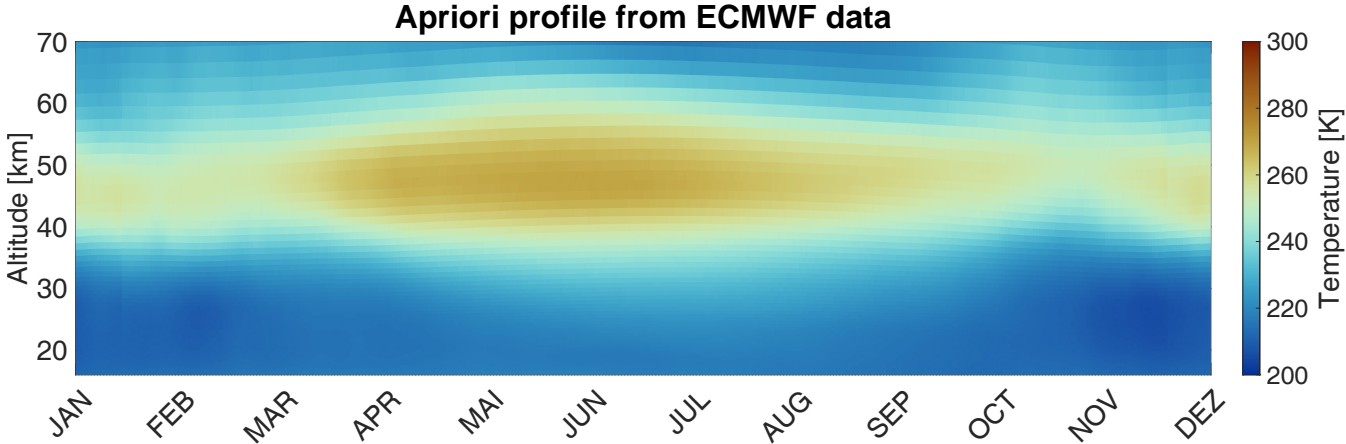

**Figure 2.** Averaged ECMWF temperature profiles for the geolocation of Payerne (CH). A moving window of 31 days was used for smoothing
Afterwards the average over the 4 years 2014-2017 was taken.

## 5.3    Tropospheric correction

The new retrieval still incorporates a tropospheric correction. The received signal is the integral along the line of sight of
all emitted microwave radiation including also tropospheric altitudes. However, the main goal of the new retrieval is the
improvement of the stratospheric and mesospheric temperature soundings, which requires a higher frequency resolution in the
line center at the cost of the much broader tropospheric signal, which is still dominating the overall brightness temperature in the
line wings and the center. Therefore, the tropospheric signal is separated and removed from the stratospheric and mesospheric
intensities by implementing a tropospheric correction. The method is based on the assumption that the troposphere can be





approximated than a homogeneous layer with a weighted mean brightness temperature;

$$T_m(\nu) = \frac{\int_{z_1}^{z_t} T(z,\nu)\alpha(z,\nu)e^{-\tau(z,\nu)}dz}{\int_{z_1}^{zt}\alpha(z,\nu)e^{-\tau(z,\nu)}dz} \quad . \tag{6}$$

Where the integral is taken from the ground $z_1$ to the top of the troposphere $z_t$, $\nu$ is the frequency, $\alpha$ denotes the absorbing coefficient, $\tau$ the opacity, and $T(z)$ is the physical thermal equilibrium temperature. The weighted mean temperature is used to estimate a mean tropospheric opacity $\tau_{trop}(\nu)$. After estimating all these parameters the brightness temperature on the top of the troposphere is determined by solving the radiative transfer equation. The integrals above are dominated by the lowest altitudes because $\alpha$ is pressure dependent and rapidly decreases with increasing altitude. Assuming a linear relationship between the surface temperature $T_s$ and $T_m$ can be expressed by;

$$T_m = aT_s + b \quad . \tag{7}$$

To determine the coefficients $a$ and $b$ radiosonde measurements at Payerne launched from MeteoSwiss were used. The coefficients for the TEMPERA frequency range are found in Navas-Guzmán et al. (2015) and take values for $a = 0.8159$ and $b = 47.211$. Further details about the method are described in Ingold and Kämpfer (1998). All previous studies based on TEMPERA have applied such a tropospheric correction (Stähli et al., 2013; Navas-Guzmán et al., 2015, 2017). Although, hitherto observations with TEMPERA indicate that the tropospheric correction seems to work well, it represents a coarse approximation that is worth to be further investigated for various weather conditions. In particular, tropospheric inversion layers might have a more crucial impact on the mean tropospheric opacity $\tau_{trop}(\nu)$.

## 5.4 Measurement errors

Statistical measurement-errors arise from two sources. The first error source is the receiver noise and the second one is atmospheric noise, which originates from fluctuations and turbulent processes in the field of view. Typically reeceiver and atmospheric noise is considered as zero mean Gaussian random process. Both together, measurement-noise-variance and atmospheric-noise-variance contribute to the measurement-error-covariance (Rodgers, 2000). Other systematic errors such as a systematic frequency shift in the channels, are often hard to be identified and, thus, are not taken into account.

In the following we briefly discuss how the measurement errors are obtained. Considering $y_{ij} = y(\nu_i, t_j)$ as the measurement matrix where $\nu_i$ is the frequency of channel number $i$ and $t_j$ is the time of spectrum number $j$ in a time series with $N$ spectra. The channels are assumed to be uncorrelated with the variance $\sigma_i^2$. The final measurement spectrum is the mean of $y_{ij}$ over time $\bar{y}_i = \frac{1}{N}\sum_j y_{ij}$, so that the variance $\bar{\sigma}_i^2$ of $\bar{y}_i$ is related to $\sigma_i^2$ as;

$$\bar{\sigma}_i^2 = \frac{1}{N}\sigma_i^2. \tag{8}$$

From this, one can calculate $\bar{\sigma}_i^2$ by taking the sample variance of $\bar{y}_i$. A more stable method is to consider the variance $\sigma_{\Delta i}^2$ of the differences $\Delta y_{ij} = y_{ij} - y_{i+1j}$ which is related to $\sigma_i$ as $\sigma_{\Delta i}^2 = 2\sigma_i^2$ and, hence;

$$\bar{\sigma}_i^2 = \frac{1}{2N}\sigma_{\Delta i}^2. \tag{9}$$





Assuming that all channels are uncorrelated, the measurement error covariance matrix $\boldsymbol{S}_\epsilon$ takes a diagonal form with entries;

$$(\boldsymbol{S}_\epsilon)_{ii} = \bar{\sigma}_i^2. \tag{10}$$

## 5.5 Apriori covariance

The apriori covariance determines the uncertainty of the apriori state. For temperature profiles one usually chooses, a constant value for each grid point and a with distance exponentially decreasing correlation. However, varying the apriori covariance with altitude can improve the retrieval significantly concerning the obtained measurement response. Since the platform altitude

was set at 12 km (see tropospheric correction), lower altitudes have to be excluded from the inversion. For this purpose, the apriori covariance $\sigma_a(z_i)$ was set to 0.1K up to 12 km ($z_i$ is the altitude at the i-th grid-point). Above the virtual platform altitude, the apriori covariance increases linearly with altitude to a value of 6 K at 50 km, and higher up in the atmosphere the value increases to 8 K at 60 km and beyond that height the covariance reaches 12 K at 70 km altitude. A linear increase with altitude avoids numerical oscillations due to sharp "jumps" in the profile, which would occur when a step function is

implemented instead. The larger values at upper altitudes of the retrieval domain are beneficial to optimize the information content of the measurement vector. However, we have to note, that this method tends to be more prone to generate some unwanted numerical effects such as spurious oscillations. On the other hand, a smaller choice of the apriori covariance for these altitudes forces the retrieval to stay close to the apriori state, and, thus, information content would be lost. The values described above were optimized through empirical tests prioritizing an optimal balance between numerical stability and high

sensitivity of the solution at the stratosphere and lower mesosphere. Considering these aspects, the covariance matrix $\boldsymbol{S}_a$ takes the form;

$$(\boldsymbol{S}_a)_{ij} = \sigma_a(z_i)\sigma_a(z_j)\exp\left(-\frac{|z_i - z_j|}{h}\right) \tag{11}$$

where $h$ is the correlation length, which was set to be $h = 1$ km.

## 5.6 Other sources of uncertainty

The advantage of the optimal estimation implementation of the retrieval is the possibility to derive the information gain from the observations (Shannon, 1948; Shannon and Weaver, 1949). An important quantity, that is widely used for error analysis in information theory, is the gain matrix given by;

$$\boldsymbol{G}_y = \frac{\partial R}{\partial \boldsymbol{y}}. \tag{12}$$

The gain matrix can be interpreted as the sensitivity of the retrieval $R$ to the measurement $\boldsymbol{y}$. Furthermore, the gain matrix can

be used to define the averaging kernel matrix by;

$$\boldsymbol{A} = \boldsymbol{G}_y \boldsymbol{K}_x. \tag{13}$$

According to Rodgers (2000) the averaging kernel is the sensitivity of the retrieval to the (unknown) true state. The rows of $\boldsymbol{A}$ provide correlations and a distinct maximum, which defines the altitude of maximum measurement response for a vertical grid





point. Their half-width can be regarded as a measure of the effective vertical resolution. The measurement response vector $\boldsymbol{mr}$
is defined as (Rodgers, 2000; Eriksson et al., 2005);

$$\boldsymbol{mr}_i = \frac{\boldsymbol{A}_i \boldsymbol{x}_a}{\boldsymbol{x}_{ai}} \tag{14}$$

where $\boldsymbol{A}_i$ is row $i$ of the averaging kernel matrix and $\boldsymbol{x}_{ai}$ is the $ith$-entry of the apriori state. For an ideal retrieval, the values
of $\boldsymbol{mr}$ equals 1. The lower the measurement response, the more apriori information is included in the solution. Measurement
responses below 0.6 indicate that the retrieved state depends mostly on our apriori information.

Weighting the measurement-error covariance matrix $S_\epsilon$ with $\boldsymbol{G}_y$ one gets the retrieval noise (or observational error) covariance
matrix

$$\boldsymbol{S}_o = \boldsymbol{G}_y \boldsymbol{S}_\epsilon \boldsymbol{G}_y^\mathsf{T}. \tag{15}$$

Another indicator is the modeling error $\boldsymbol{s}_M$, obtained by weighting the retrieval residuals with $\boldsymbol{G}_y$

$$\boldsymbol{s}_M = \boldsymbol{G}\left[\boldsymbol{y} - F(\boldsymbol{x}, \boldsymbol{b})\right]. \tag{16}$$

In theory, this vector should be evaluated at the true state $\boldsymbol{x}$ and $\boldsymbol{b}$ which is, of course, not known. Evaluating this quantity at
the retrieved state $\hat{\boldsymbol{x}}$ instead will lead to slightly increased values.

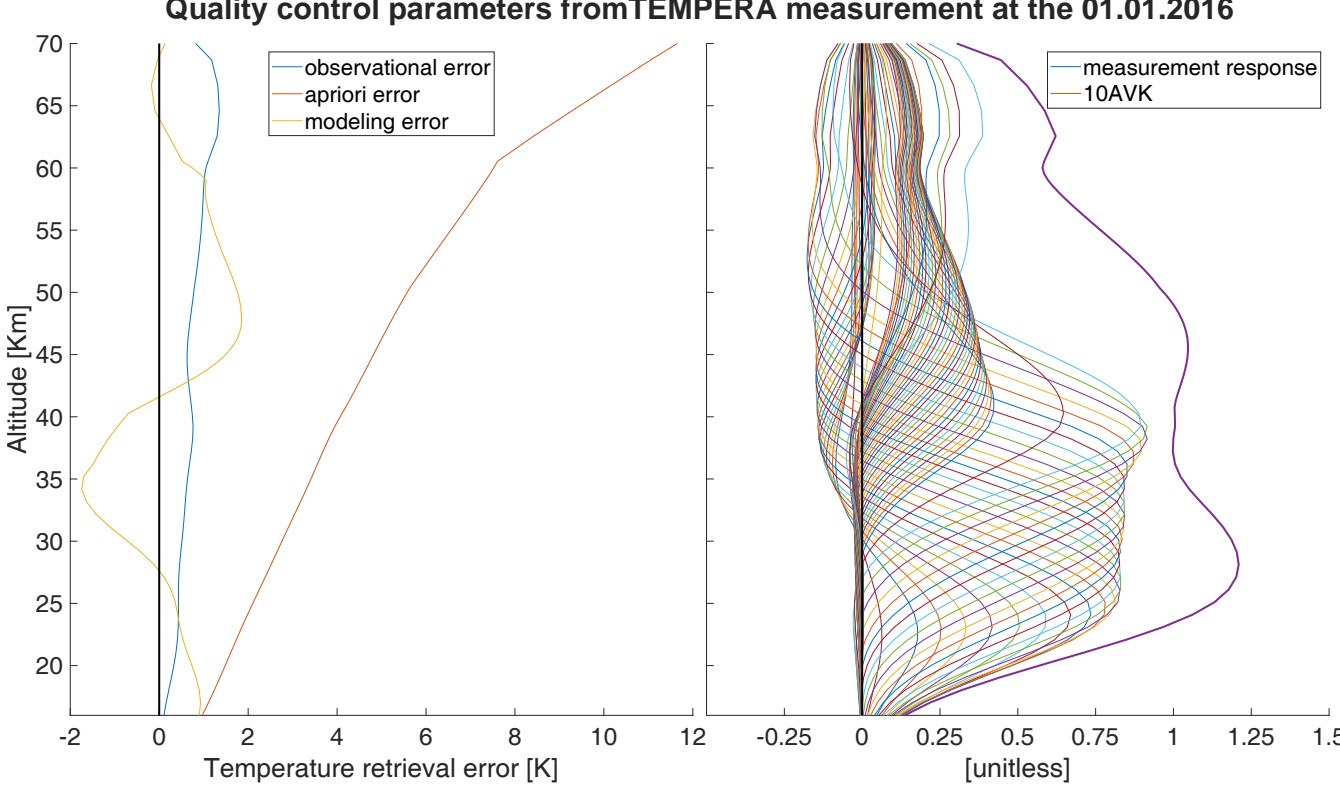

**Figure 3.** Different error components from the retrieval method (left). Units of errors are always Kelvin here.
Measurement response (MR) and Averaging kernel matrix (AVK) (right). The AVK is multiplied by a factor of 10 for a facilitated comparison
with the MR.

As an example, a set of quality control parameters are illustrated in Figure 3. The modelling error, which is directly related
to the forward model residuum, leads to the conclusion that the retrieved profile is underestimated around 30-40km and over-
estimated around 45-55 km by about 2 K.

The AVK up to 40 km shows the expected and already documented behaviour (Stähli et al., 2013; Navas-Guzmán et al., 2017),
where the best performance is reached at an altitude around 30 km. From 40 km upwards the Zeeman calculation leads to a
second but lower peak between 40-50 km. The last peak between 60-65 km is due to the increased apriori error in this region.
This behaviour is also reflected in the measurement response. As a rule of thumb, the altitude range of a retrieved profile is
usually defined as the region where the measurement response is above 0.8. In this plot, this region is between 22-53 km.

**6   Temperature retrievals including Zeeman effect**

The revised temperature retrieval was applied to data collected with TEMPERA in Payerne between 2014-2017. The main
differences compared to previous works from Navas-Guzmán et al. (2015, 2017) is the inclusion of Zeeman effect in the center



of the oxygen emission lines, and the use of updated apriori and measurement covariances to improve numerical stability and the retrieval sensitivity. Furthermore, the new retrieval emphasis on stratospheric and mesospheric altitudes to observe tidal

waves and their temporal intermittency. The temporal resolution was slightly decreased from 2 hours (Navas-Guzmán et al., 2017) to about 2.5 hours. The increased integration time resulted in more robust temperature estimates. On average we obtained 8-9 integrated spectra per day. Each integrated spectra consists of about 150-160 individual atmospheric soundings/spectra obtained from atmospheric observations lasting 0.5 seconds (mirrow pointing towards sky) using the stratospheric/mesospheric measurement mode with the high-resolution FFT-spectrometer.

We also implemented a quality control before the averaged spectra is computed. Some spectra are removed from the averaging due to increased atmospheric noise, mainly caused by tropospheric weather e.g., strong precipitation or temporary technical issues with the instrument. On average about 3.6% of the integrated spectra are removed from the analysis within the 4 years of observations.

Figure 4 shows temperature soundings for TEMPERA, MERRA2 and NAVGEM-HA for the whole period (2014-2017). The

seasonal pattern indicates higher temperatures at all retrieved altitudes during the summer season and lower temperatures at the stratosphere during the winter months. The winter months are characterized by an increased planetary wave activity and the frequent occurrence of sudden-stratospheric-warmings (Scherhag, 1952; Matsuno, 1971; Limpasuvan et al., 2016; Matthias et al., 2013). Also the spring transition is clearly distinguishable from the temperature data (Matthias et al., 2021).

A first comparison of the retrievals with the apriori profile is shown in Figure 5. At altitudes from 20-35 km and between 45-60

km TEMPERA observes warmer temperatures compared to the apriori, whereas between 35-45 km TEMPERA often yields lower temperatures. Furthermore, from 60 km and higher up, the colors are brighter indicating the decrease in measurement response consistent with the averaging kernels presented in Fig.3 for these altitudes.





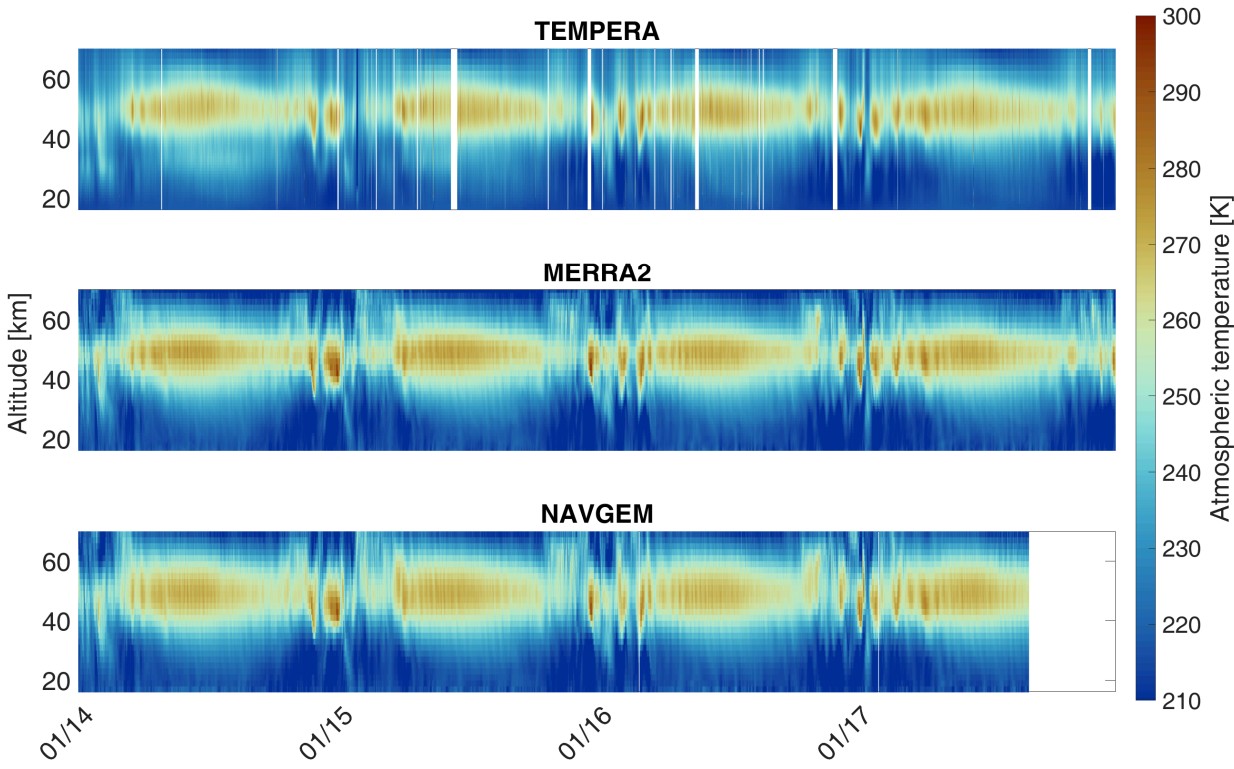

**Figure 4.** Continuous atmospheric temperature profiles, retrieved from TEMPERA measurements in comparison to MERRA2, and NAVGEM-HA data for the years 2014-2017 over the geolocation of Bern (CH). The altitude range is 53 km. Above 53 km the retrieved profiles are dominated by the apriori profiles.

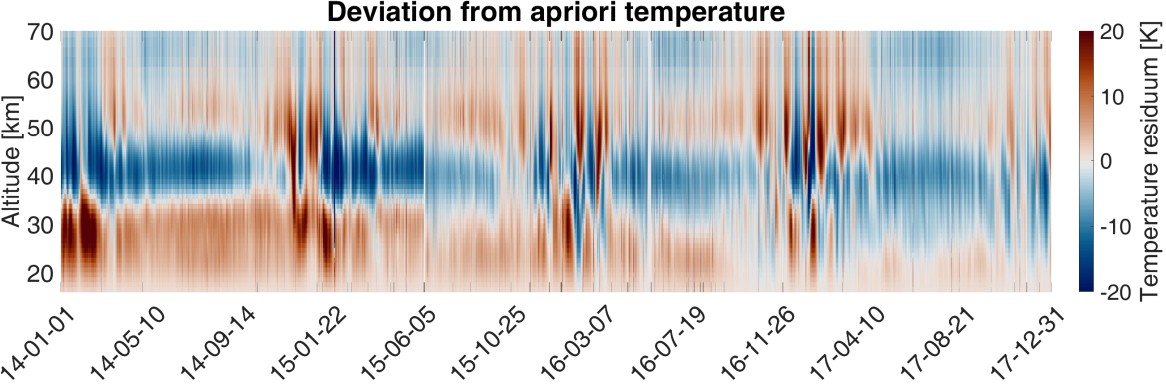

**Figure 5.** Absolute differences between retrieved TEMPERA profiles and apriori profiles. Reddish regions indicates higher values for the retrieved quantities in comparison to the apriori. Bluish areas support colder temperatures concerning the apriori state.



# 7 Comparison of temperature retrievals to MERRA2 and NAVGEM-HA

The performance of the new temperature retrievals is assessed by comparing our observations to state-of-art reanalysis data

from MERRA2 and the meteorological analysis of NAVGEM-HA. Therefore, we compute correlation coefficients based on monthly medians and corresponding variances for all data sets. These monthly medians essentially remove all atmospheric waves on short time scales such as tides and gravity waves from the model fields as well as the temperature soundings. However, we have to note that atmospheric time series cannot necessarily be considered as Gaussian random variables. Often the atmospheric natural variability exceeds the statistical uncertainty of the observations (e.g., Stober et al., 2017, see Figure

3) and, thus, an overestimation or inflation of the correlation coefficients is the result. Assuming a linear regression model between the TEMPERA profiles $T_{TMP}(z)$ and the profile for cross-comparison $T_{CCP}(z)$

$$T_{TMP}(z) = m T_{CCP}(z) + q,\tag{17}$$

the coefficients $m, q$ were determined through linear regression. For two statistically identical data sets, we would obtain $m = 1$, and $q = 0$. The coefficient $q$ gives an absolute offset of the two profiles, while a slope $m$ above 1 indicates a higher

sensibility of the profile $T_{TMP}(z)$ (see section 8) relative to the compared profile $T_{CCP}(z)$. This method gives a quantitative estimation of the absolute offset but provides no information at which altitude this occurs. In Figure 6 and Figure 7 we show linear correlation coefficients of the median monthly temperature profiles for the year 2016 for TEMPERA vs MERRA2 and TEMPERA vs. NAVGEM-HA, respectively. The error bars correspond to the temperature variance for each data set. The correlations are estimated after subtracting the median temperature from each profile, which was estimated to be approximately

250 K

$$T_{TMP} \rightarrow T_{TMP} - 250 \text{ K}$$
$$T_{CCP} \rightarrow T_{CCP} - 250 \text{ K}.\tag{18}$$

$$\tag{19}$$

The shift of the temperature profile to lower values is necessary because the linear regression would otherwise falsely give

good values for $m$. The other years can be found in appendix A.





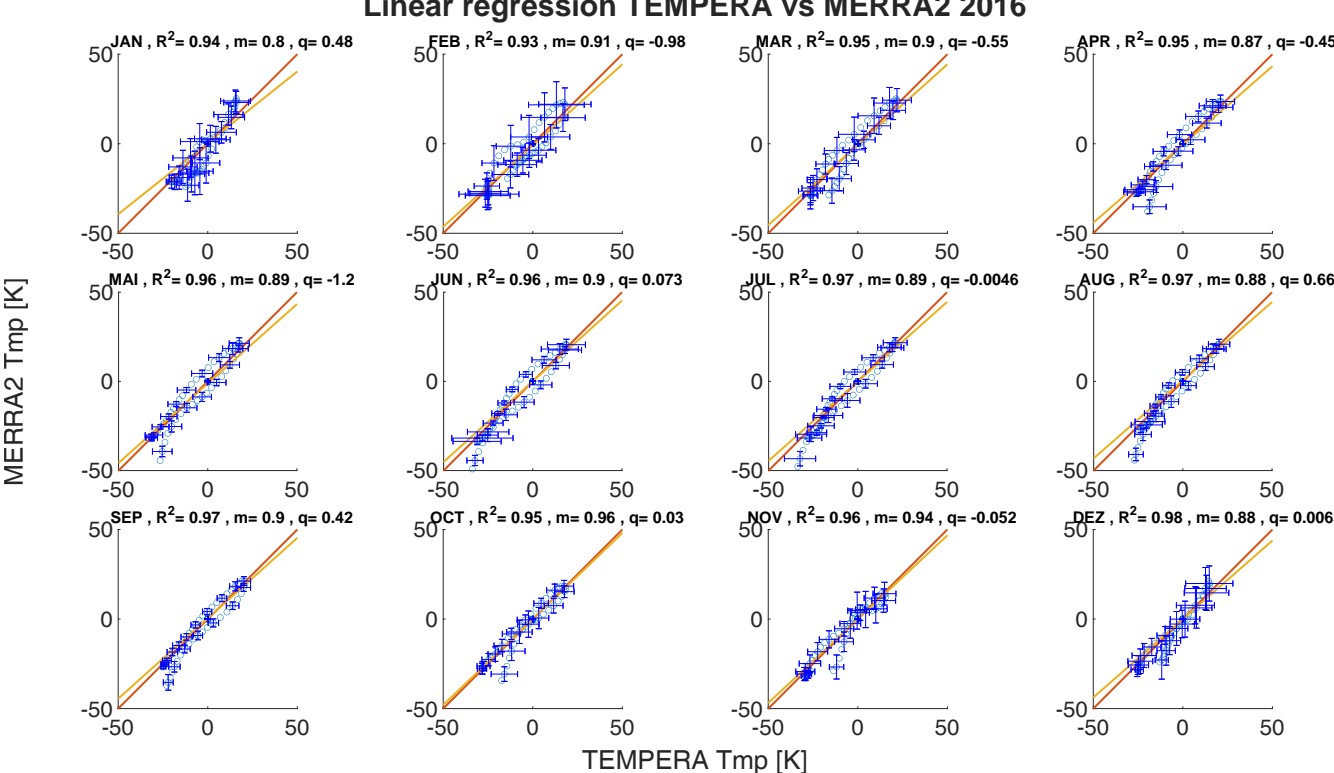

**Figure 6.** Linear Regression of TEMPERA against MERRA2 temperatures.



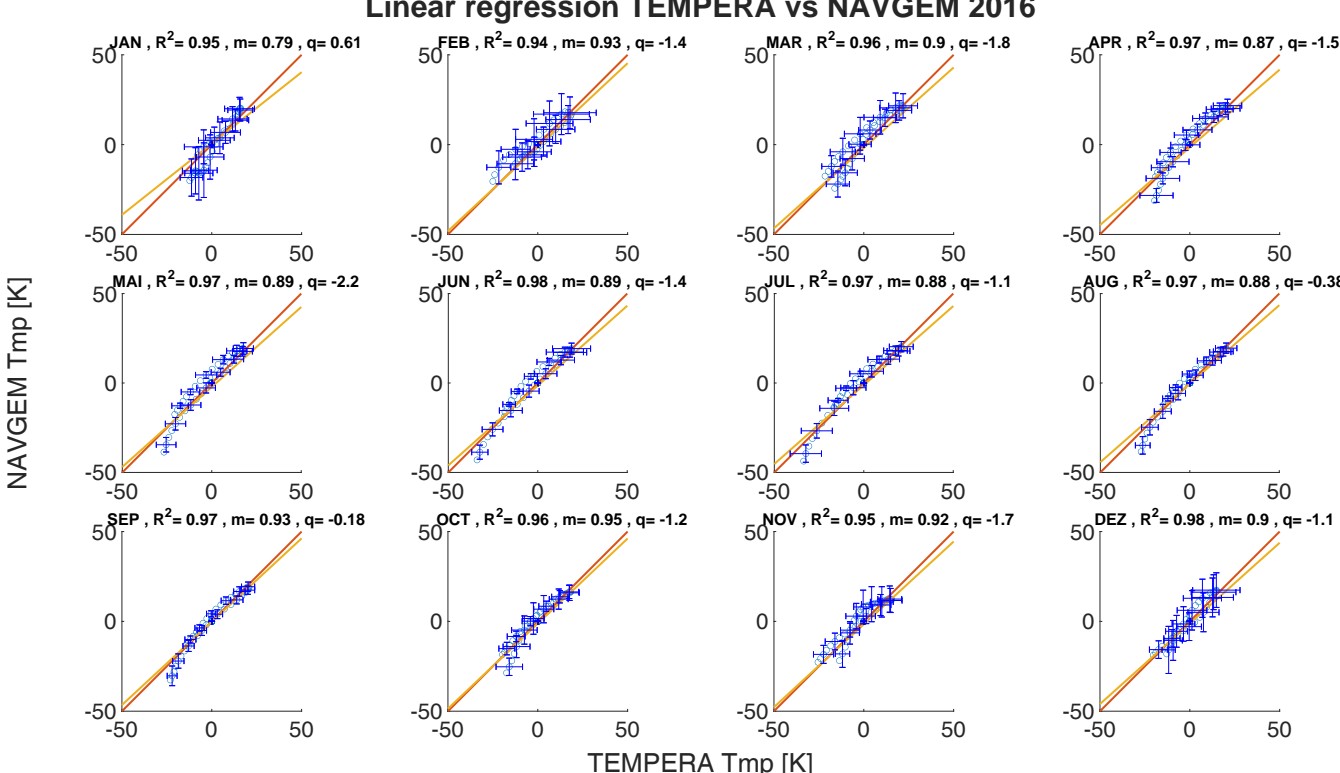

**Figure 7.** Linear Regression of TEMPERA against NAVGEM-HA temperatures.

The monthly median temperature correlation coefficients exhibit a range between 0.93-0.98 for the comparison with MERRA2 and about 0.94-0.98 for NAVGEM-HA. The highest correlation coefficients are achieved during the summer months from April to September and in December. The lowest correlations are found during January and February and are the result of the increased planetary wave activity and the more variable polar vortex dynamics in 2016 (Matthias et al., 2016; Stober et al., 2017; Matthias and Ern, 2018). NAVGEM-HA indicates a similar seasonal behaviour for the year 2016 and occasionally has minimal larger correlations. The mean temperature bias $|q|$ between the new TEMPERA retrievals and MERRA2 is smaller than 1.5 K. The temperature bias relative to NAVGEM-HA takes values between -0.1 K up to -2.2 K (excluding the exceptional January 2016). The slopes $m$ of the linear regression with MERRA2 and NAVGEM-HA are in a range between $m = 0.8$ and $m = 0.96$ indicating a lower sensitivity of TEMPERA to the atmospheric variability relative to the model fields. We also estimated yearly median altitude resolved Pearson correlation coefficients. These are shown in Figure 8. It is remarkable that the correlation coefficients are most of the time larger than 0.8 and often exceed 0.9 for MERRA2 and NAVGEM-HA, respectively. Furthermore, the seasonal correlation coefficients reveal a sharp drop off at about 53-55 km, which appears to be the limiting altitude for TEMPERA temperatures and the new retrieval. Above this altitude, the solutions of the retrieval are dominated by apriori information. The comparison also indicates that NAVGEM-HA seems to show a slightly higher correlation concerning TEMPERA temperatures compared to MERRA2.





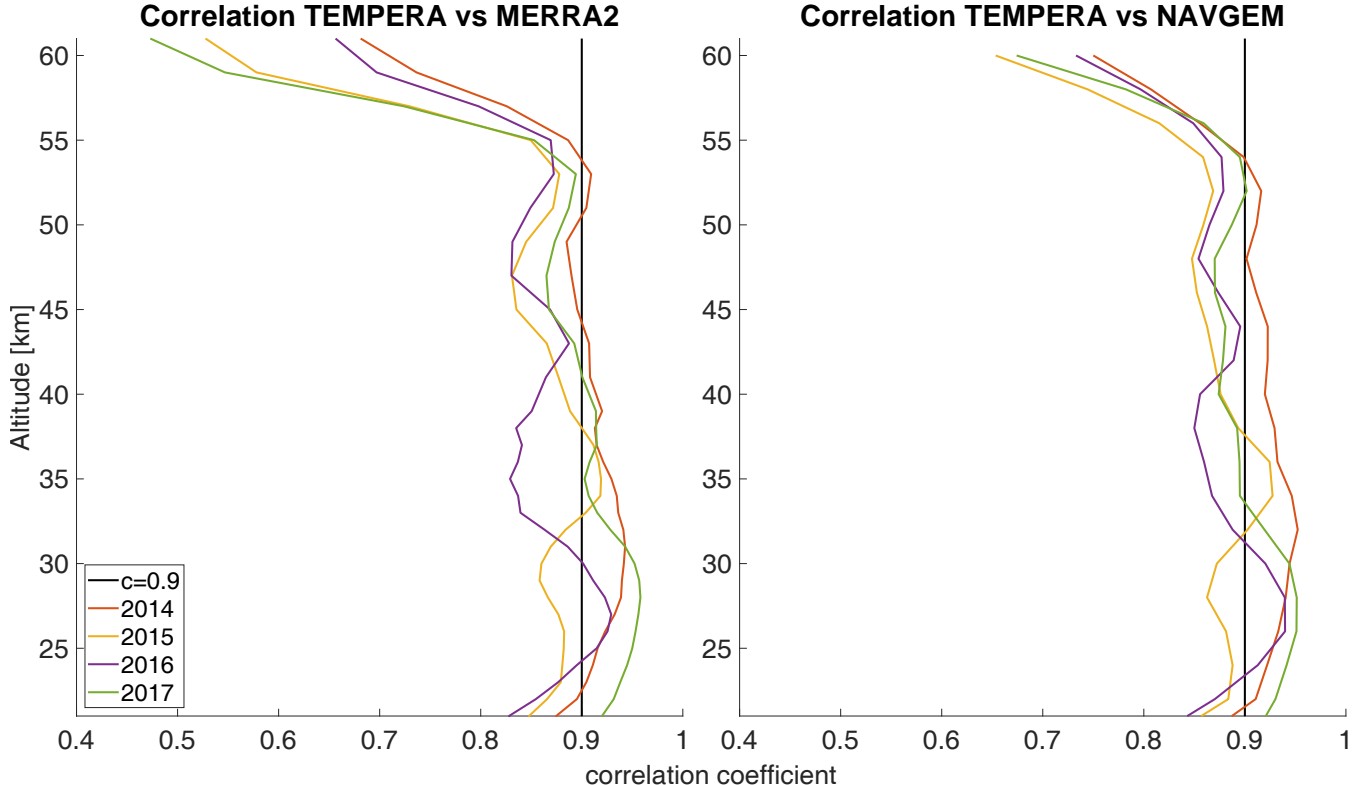

**Figure 8.** Correlation coefficients over altitude between TEMPERA and MERRA2 data, and TEMPERA and NAVGEM-HA data.

Another important aspect to compare are altitude-time dependent systematic differences between MERRA2 and NAVGEM-HA. Therefore, we compute altitude time residuals by subtracting MERRA2 and NAVGEM-HA from the temperatures observed by TEMPERA. Figure 9 shows the resulting temperature residuals for both model data sets and the complete time series. Similar to the Pearson correlation coefficients MERRA2 and NAVGEM-HA reflect the same characteristic systematic differences. Furthermore, the installation and upgrade of the receiver around 5th June 2015 is clearly visible in the residual comparison. The new receiver reduced the standing wave contamination in the line wings and, thus, mostly affected the data quality below 40 km altitude.

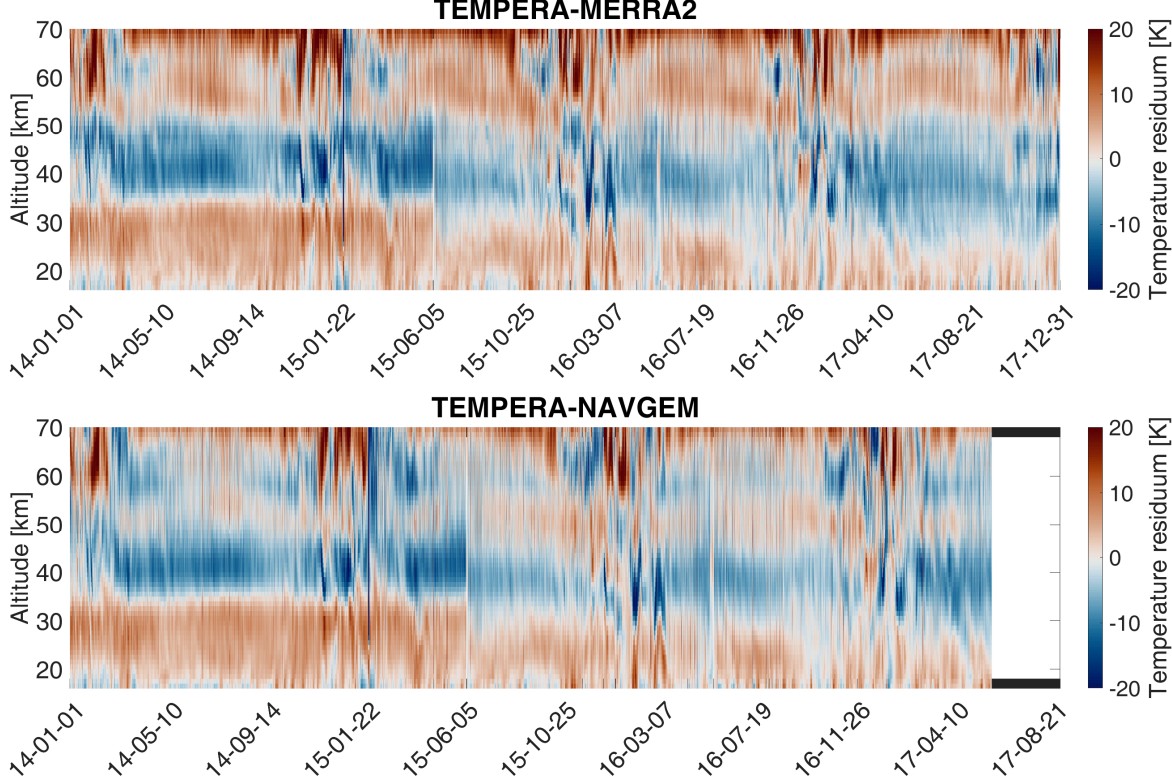

**Figure 9.** Absolute differences between TEMPERA and MERRA2 (upper panel), and NAVGEM (lower panel). Red regions indicates higher values of TEMPERA.

The vertical residuals show some systematic and altitude dependent differences. Below 35 km there is a tendency that TEMPERA shows warmer temperatures compared to MERRA2 and NVAGEM-HA. Between 35-50, the models seem to have a warm bias compared to the radiometric temperature sounding. Above 53 km MERRA2 indicates a clear tendency to underestimate the temperatures relative to TEMPERA, whereas NAVGEM-HA shows a more variable vertical structure of the residual temperatures exhibiting times and altitudes with warmer, but also periods and heights with colder temperatures. It is also evident from the residual comparison that during the winter season the increased planetary wave activity leads to larger differences between our temperature observations and the model data.

Finally, Figure 10 presents a comparison of the 3 hourly resolved temperature time series at 50 km altitude. The comparison underlines that the TEMPERA observations exhibit still such a high measurement response at this height that the temperature amplitude and phase of planetary waves is well-captured in comparison to MERRA2 and NAVGEM-HA. There is also a characteristic diurnal tidal oscillation in MERRA2, NAVGEM-HA and the radiometer data visible. Overall the measurements from TEMPERA and the MERRA2 and NAVGEM-HA temperature agree within a few kelvin (5-10 K). Furthermore, the comparison supports that it is feasible to obtain tidal information from the TEMPERA temperature soundings on a daily basis.



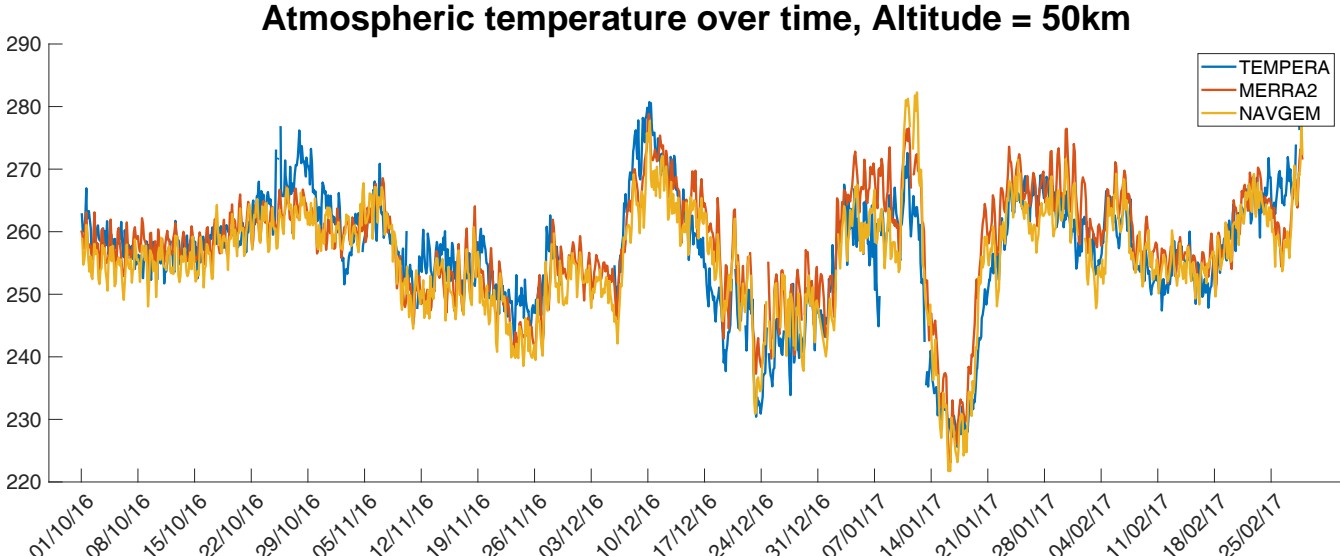

**Figure 10.** Comparison of TEMPERA, MERRA2, and NAVGEM temperatures on larger timescale (months), at which planetary waves occurs and smaller timescale (days), which is the timescale of atmospheric tides.

## 8 Discussion

The main goal of the new retrieval algorithm was the implementation of the Zeeman effect in the temperature retrievals, which was not available in previous versions of the radiative transfer model for both oxygen emission lines. Thus, the new temperature retrieval yields an increased measurement response and altitude coverage up to 55 km compared to former TEMPERA

observations where 45-48 km seemed to be the limiting altitude (Stähli et al., 2013; Navas-Guzmán et al., 2015). Furthermore, the new retrieval was optimized concerning the apriori state vector and covariances, which also led to some improvement at the upper stratosphere and lower mesospheric heights. TEMPERA observations offer the possibility to perform continuous temperature measurements at altitudes between 16-55 km.

While implementing a new retrieval method it is always necessary to achieve a balance between numerical stability and sen-

sitivity to the atmospheric state. A small apriori covariance or a too large measurement error results in low sensitivity of the retrieval, although such a retrieval is stable concerning numerical oscillations. On the other side, such a retrieval likely underestimates the natural or true variability of the estimated parameters. The solution tied to the apriori state. A large apriori covariance improves the sensitivity of the retrieval, but at the cost of numerical oscillations, which can dominate the whole retrieved profile. The new retrieval is well-balanced to achieve the highest possible sensitivity at 50 km while avoiding numerical

instabilities/oscillations.

Statistical measurement errors are known with high precision, the final error on the temperature profile is rather a measure of the information content than an error in the classical sense. The state of the art method is a cross-comparison of different and independent data sets. Calculations of correlation coefficients or goodness-of-fit values ($R^2$) in a linear regression requires





always some information about the uncertainty of the data set. Since this information is missed, usually the sample variation
is taken instead. This approach should however be used with appropriate caution because atmospheric profiles or time series
are not random variables and natural variations could be bigger than the actual errors. This circumstance leads directly to an
overestimation of the correlation coefficients of two compared data sets.

Continuous temperature observations at the stratosphere and mesosphere are rare. Lidars are often limited by the tropospheric
weather conditions and only a few long observations are available (e.g., Stober et al., 2017; Baumgarten and Stober, 2019; Eix-
mann et al., 2020). However, these lidar studies underline that continuous temperature observations are essential to investigate
atmospheric wave and their intermittency covering periods from gravity waves, tides and planetary waves at the source region,
to research wave-wave interactions.

Satellite observations from MLS or SABER provide neither the temporal and spatial resolution to resolve all atmospheric
waves and their intermittent behaviour. Due to the spacecraft orbit and viewing geometry very often only one measurement
per day is available for a certain geographic location. However, satellite observations are a key information source for data
assimilation into MERRA2 and NAVGEM-HA at the stratosphere and mesosphere for the temperature and dynamical fields
(Gelaro et al., 2017; Kuhl et al., 2013; Eckermann et al., 2018). Other meteorological observations such as radiosondes reach
only altitudes of about 28-38 km and, thus, provide only temperature, wind or chemical information at the lower and middle
stratosphere. Furthermore, radiosondes are launched every 12 hours or at some stations occasionally every 6 hours, which
limits their impact to capture atmospheric tides at the stratosphere.

Navas-Guzmán et al. (2017) already performed an intercomparison of the TEMPERA observations with MLS satellite data,
lidar and radiosondes and WACCM. The obtained Pearson correlation coefficients were between 0.9 to 0.94 for a 3-year-
long time series that were interpolated to match the different temporal resolutions and emphasized altitudes between 22-43 km
where the measurement response was larger than 0.8. In this study, we already achieved this degree of correlation using median
monthly profiles and for yearly observations for the altitude range from 20-55 km. However, the comparison to MERRA2 and
NAVGEM-HA still exhibits a warm bias of TEMPERA for the altitude range between 20-30/35 km and a cold bias between
30(35)-48 km, which was already found in Navas-Guzmán et al. (2017). Some of the systematic biases at the lower altitudes
as well as at the upper altitudes occur at heights with a low measurement response and thus, we investigated a potential apriori
dependence by computing similar climatologies for MERRA2 and NAVGEM-HA as shown in Figure 2 for ECMWF. A com-
parison of these a priori climatologies between the all three reanalysis data sets revealed similar altitude dependent offsets than
the TEMPERA comparison and explains most of the upper stratospheric bias and at least partly the lower stratospheric offset.

## 9 Conclusions

In this study, we reprocessed observations of the TEMPERA radiometer conducted between 2014 until 2017 with a recently
developed and updated temperature retrieval. The new algorithm accounts for the Zeeman effect in the line center for both
oxygen emission lines and uses revised apriori information for the state vector and covariances. We demonstrate with the new



retrievals that TEMPERA temperature soundings can be carried out nearly continuous and with an increased altitude coverage by leveraging the updated radiative transfer model (ARTS) and HITRAN quantum numbers, which were not available previously.

We validated the retrieved temperature against the MERRA2 reanalysis and the meteorological analysis NAVGEM-HA for the years 2014-2017. Seasonal Person correlations coefficients remained between 0.85-0.95 between 20-55 km altitude. Therefore, we conclude that considering the Zeeman effect in the line center together with the revised apriori information resulted in an extended altitude coverage of about 8-10 km compared to the previous algorithm applied to the same TEMPERA measurements while sustaining the temporal resolution (Stähli et al., 2013; Navas-Guzmán et al., 2017).

Furthermore, we assessed the correlation coefficients and mean biases for monthly median temperature profiles of TEMPERA and the validation data MERRA2 and NAVGEM-HA. We obtained correlation values between 0.8-0.96 throughout the course of the year. The smallest correlations are found in January and February during strong planetary wave activity or for stratospheric warming evens. The summer months from April to September reached correlations between 0.94-0.96. The mean temperature bias between MERRA2 and the radiometric temperatures was smaller than $\pm 1K$ and basically vanished for some

375 months. However, the comparison to NAVGEM-HA resulted in a cold bias between 1-2 K for the TEMPERA temperatures. Altitude-dependent differences were examined by computing temperature residuals of TEMPERA and both model data sets. We identified that the lower altitudes between 20-35 km tend to exhibit a warm bias of approximately 5 K for the radiometer and from 35-50 km we found a systematic cold bias of approximately 5 K for TEMPERA compared to MERRA2 and NAVGEM-HA. Above 50 km altitude, MERRA2 and NAVGEM-HA also start to show some discrepancies in the vertical temperature

structure. During strong planetary wave activity in the winter months the differences between MERRA2, NAVGEM-HA and TEMPERA exceed $\pm 10$ K. However, it remains unclear whether these biases or differences are due to the instrument or whether the reanalysis models might produce some deviations due to the changes in the assimilated observations as the true atmospheric state remains unknown.

*Data availability.* MERRA-2 data are available at MDISC, managed by the NASA Goddard Earth Sciences (GES) Data and Information

Services Center (DISC) DOI:10.5067/QBZ6MG944HW0. TEMPERA temperatures are shared on request (gunter.stober@unibe.ch). The NAVGEM-HA data is available upon request from NRL.





## Appendix A

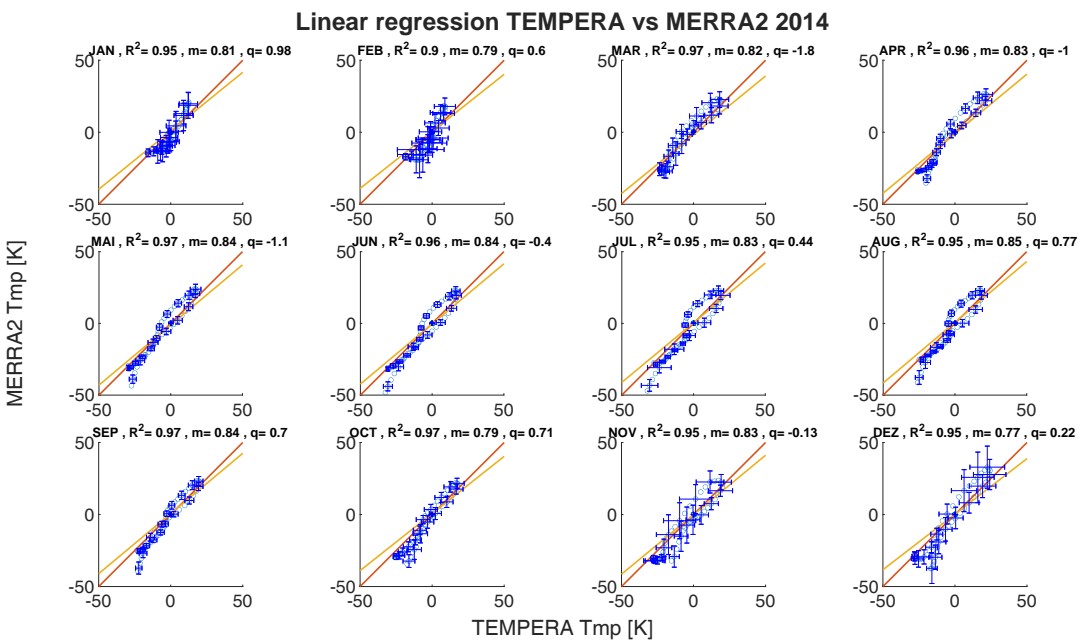

**Figure A1.** Linear Regression of TEMPERA against MERRA2 temperatures for the year 2014.

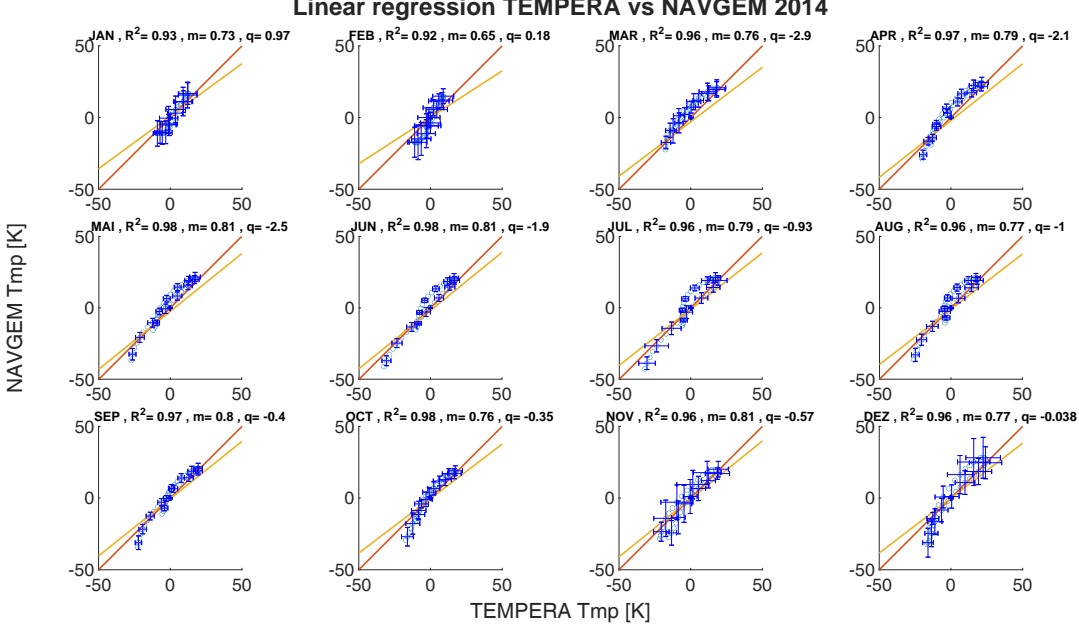

**Figure A2.** Linear Regression of TEMPERA against NAVGEM temperatures for the year 2014.



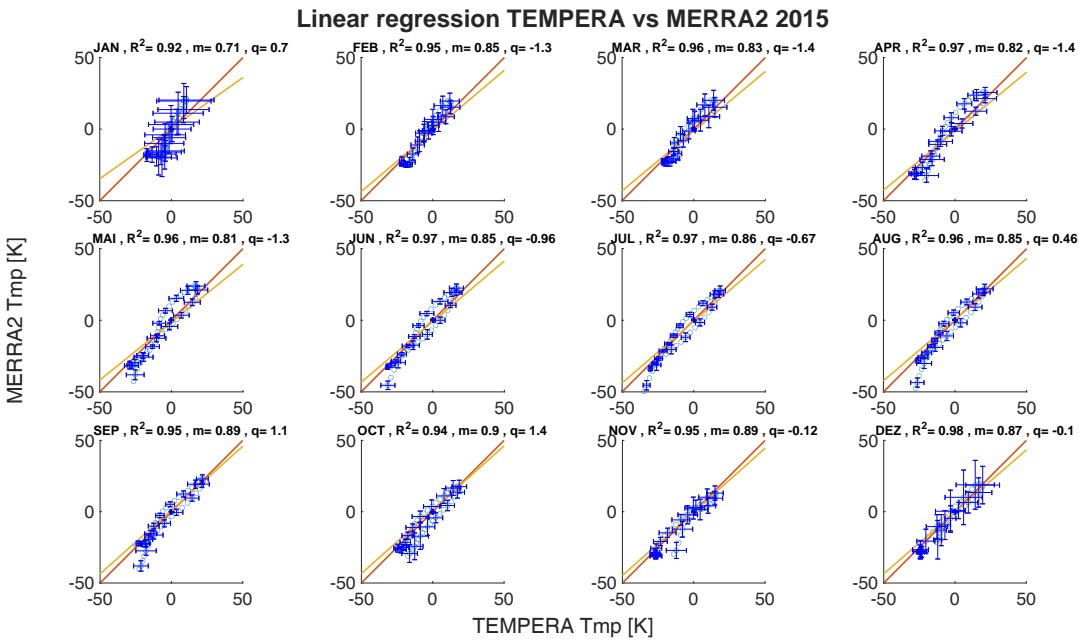

**Figure A3.** Linear Regression of TEMPERA against MERRA2 temperatures for the year 2015.

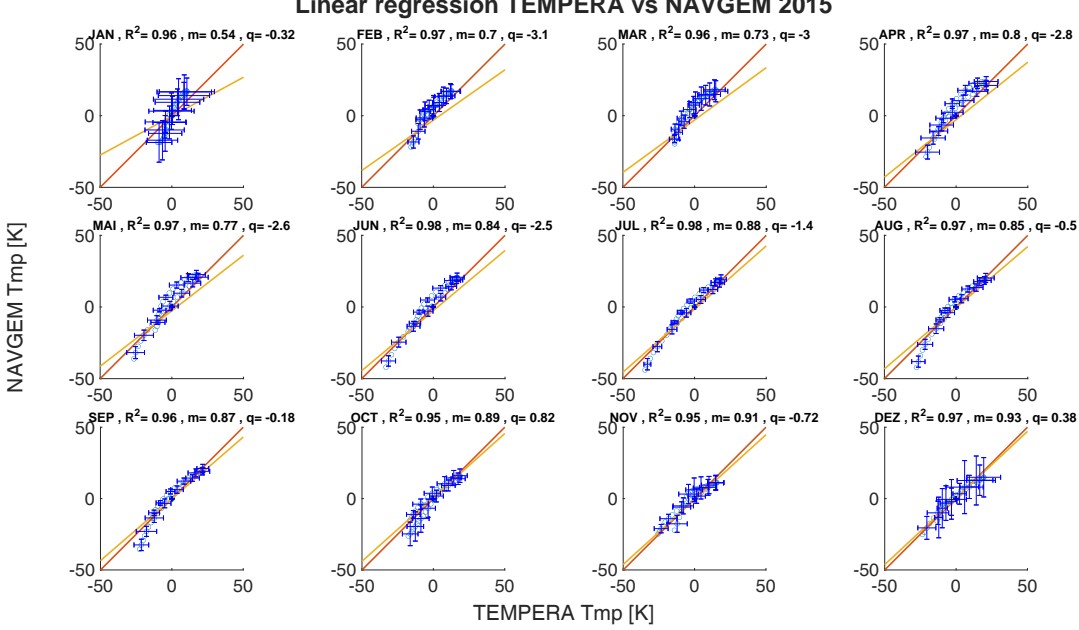

**Figure A4.** Linear Regression of TEMPERA against NAVGEM temperatures for the year 2015.





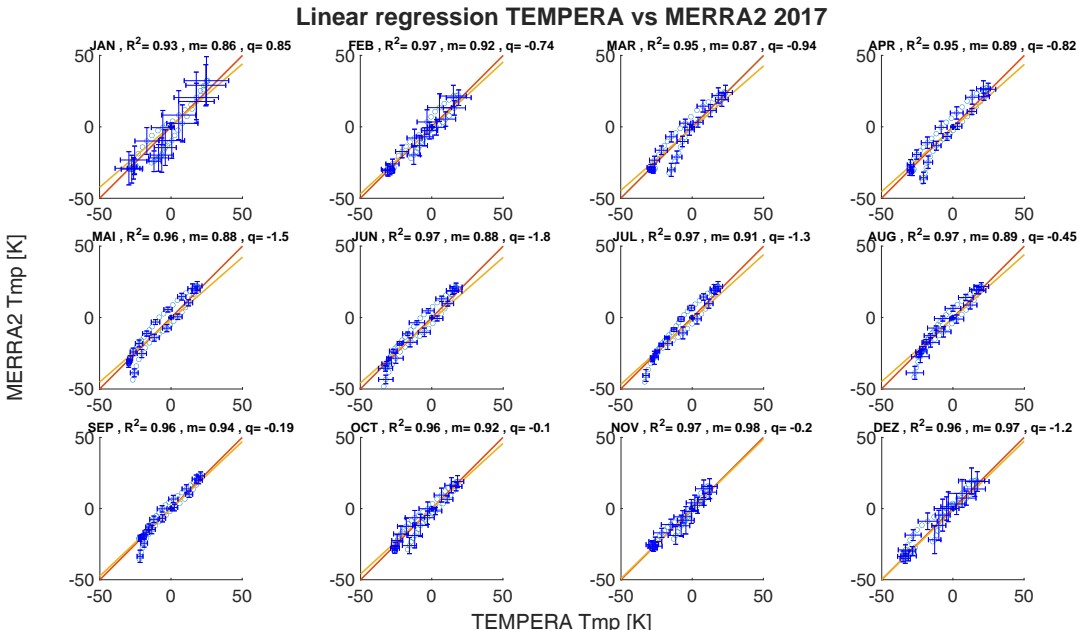

**Figure A5.** Linear Regression of TEMPERA against MERRA2 temperatures for the year 2017.

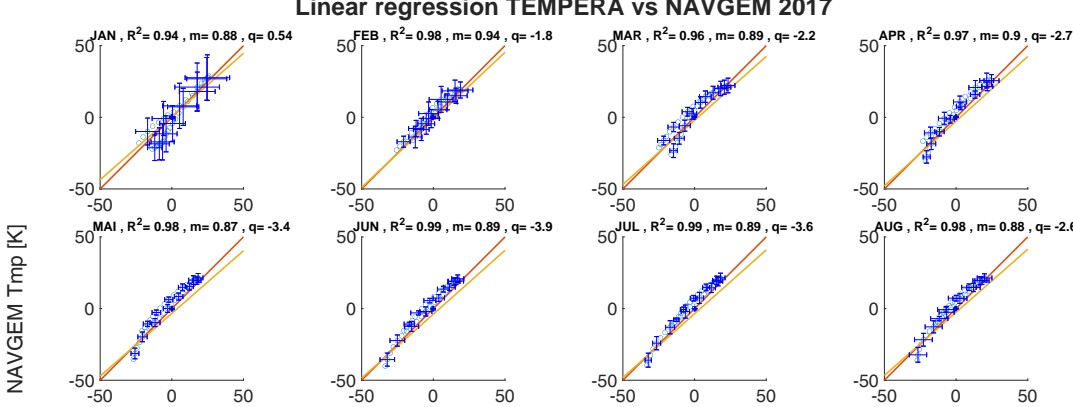

**Figure A6.** Linear Regression of TEMPERA against NAVGE temperatures for the year 2017.



*Author contributions.* WK and GS conceptualized the content of the manuscript. WK implemeneed the retrieval and performed the data analysis of TEMPERA observations. GS reduced the MERRA2 and NAVGEM-HA data for the validation. AM, FN and NK guided and supported the preparation of the manuscript and developed TEMPERA. All authors contributed to the editing of the manuscript.

*Competing interests.* The authors declare that they have no competing interests.

*Acknowledgements.* The authors acknowledge the European Centre for Medium-Range Weather Forecasts (ECMWF) for the supplied data. This research has been supported by the Schweizerischer Nationalfonds zur Förderung der Wissenschaftlichen Forschung (grant no. 200021-200517 / 1), and the Swiss Polar Institute (SPI) supports the developement of the TEMPERA-C radiometer. We thank the ARTS developer team for their support implementing the Zeeman effect into the retrievals. Scientific colour maps (Crameri et al., 2020) are used in this study
to prevent visual distortion of the data and exclusion of readers with colour-vision deficiencies."





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
