# Peer review of "Continuous temperature soundings at the stratosphere and lower mesosphere with a ground-based radiometer considering the Zeeman effect"

_Atmospheric Measurement Techniques, 2021_

## Author Response (AR1)

We thank the reviewer for the constructive and helpful suggestions to improve our manuscript. We appreciated this review. There will be a revised manuscript with tracked changes and a detailed response to the raised points.

***Comment:***
*The intrusion of Zeeman line splitting is an important part of the manuscript. As you mention Navas-Guzmán et al. (2017) compared TEMPERA observations with for example MLS. I wonder why you, in this investigation, did not compare with the same instruments to see the effect of including the Zeeman effect?*

***Answer:***
*MLS data is included in the NAVGEM-HA dataset hybrid-4DVAR data assimilation. Comparisons with the MLS data would differ only slightly from them with NAVGEM-HA and we considered this as redundant information.*

***Comment:***
*You compare TEMPERA with the two reanalysis systems MERRA2 (Gelaro et al., 2017) and NAVGEM-HA (Eckermann et al., 2018), which both assimilate satellite data. I think it would be valuable to, except for comparisons with TEMPERA, also compare them to each other.*

***Answer:***
A detailed comparison between both models is beyond the scope of this paper. We added a difference figure in the appendix and refer to the differences in the assimilated data sets above 50 km for NAVGEM-HA (Eckermann et al., 2018) and MERRA2 (Gelaro et al., 2017). Apparently, this is also the altitude where both models start to show systematic deviations. However, we did not investigate whether this is caused by the change in the assimilated observations between both models or whether the model physics favors a certain state.

***Comment:***
*Finally, I think it would be interesting to run the inversion software twice, one with including the Zeeman effect and one without and compare these two runs with MERRA2 and NAVGEM-HA.*

***Answer:***
*We run the inversion algorithm a second time with deactivated Zeeman effect.*

*We added difference Plots according to Figure. 9 and yearly Pearson correlation coefficients according to Figure. 8 in the appendix. Plots of the linear regression according to Figures 6 and 7 were considered to overload the manuscript and are available on request.*

*Usually, the line center for this calculation is ignored and only the line wings are used for calculations without Zeeman effect. The resulting retrieved profiles would differ only slightly from them which were calculated with activated Zeeman effect for an altitude below 45 Km. Above 45 Km the profiles with deactivated Zeeman effect would stay on the apriori. A comparison between these profiles would not provide more insight.*

*Instead, we performed an analysis leveraging the new retrieval algorithm but without Zeeman effect keeping the spectral information at the line center. This demonstrates the impact of Zeeman broadening on the retrieved profile. Comparison of these calculations with MERRA2 and NAVGEM-HA are added in Figures 8 and 9.*